# Unilateral Uterine Artery Embolization as a Treatment for Patients with Symptomatic Fibroids—Experience in a Case Series

**DOI:** 10.3390/medicina58121732

**Published:** 2022-11-27

**Authors:** Krzysztof Pyra, Maciej Szmygin, Hanna Szmygin, Sławomir Woźniak, Tomasz Jargiełło

**Affiliations:** 1Department of Interventional Radiology and Neuroradiology, Medical University of Lublin, Jaczewskiego 8 Str., 20-090 Lublin, Poland; 2Department of Endocrinology, Medical University of Lublin, 20-090 Lublin, Poland; 33rd Department of Gynecology, Medical University of Lublin, 20-090 Lublin, Poland

**Keywords:** fibroids, uterine artery embolization, unilateral, long-term results

## Abstract

*Background and Objectives:* Uterine artery embolization (UAE) has become an accepted and widely performed therapy for patients with symptomatic (reporting at least two of the following symptoms: severe or prolonged menstrual bleeding, abdominal pain, tension in abdomen, problems with urination, constipation or anemia) uterine fibroids. Although in the majority of cases, bilateral occlusion is required to obtain a successful clinical outcome, there are patients in whom treatment of only one uterine artery could be attempted. There are several reasons for unilateral UAE: hemodynamic conditions, technical difficulties, anatomical variants and unilateral dominancy of blood supply to the fibroid. Our aim is to present our 10-year experience with unilateral UAE and evaluate the radiological and clinical outcomes. *Materials and Methods:* Records of 369 patients with fibroids who underwent UAE from 2010 to 2021 were analyzed. We identified 26 patients treated with unilateral uterine artery embolization and analyzed the data of these patients. All patients attended medical consultation, were assessed using a five-grade symptom scale and underwent MRI examination. Clinical response was evaluated at least 6 months after the procedure and was categorized to one of the following groups: complete improvement, partial improvement, no change and a worsening in symptoms. *Results:* Twenty-two patients (85%) reported at least partial improvement 6 months following the procedure. One patient required secondary embolization due to recanalization. The secondary procedure was successful, and complete improvement was achieved. One patient did not observe any clinical improvement, and in two cases, symptom recurrence was observed. All three patients were referred for surgical treatment. No major complications were noted. Overall, the success rate was 88%. *Conclusions:* The results of our study support the statement that elective unilateral embolization is an appropriate treatment in patients with a dominant uterine artery.

## 1. Introduction

Uterine fibroids are the most common benign neoplasms of the pelvis among women in reproductive age [1]. Although they may remain asymptomatic, approximately 50% of all patients with fibroids complain about clinical symptoms, including menorrhagia, which may lead to anemia, bladder and bowel dysfunction, as well as lower back pain resulting from the mass effect of the fibroids, abdominal distension and infertility [2,3]. Traditional surgical interventions, such as hysterectomy and myomectomy, may be successfully replaced by minimally invasive endovascular embolization in selected patients. Since its first introduction nearly 30 years ago, uterine artery embolization (UAE) has established its role as a safe and effective therapy for patients with symptomatic fibroids [4,5].

Although classical endovascular intervention includes the occlusion of both uterine arteries, reports on successful unilateral embolization and its potential advantages (decrease in myometrial ischemia, lower risk of non-target embolization of the ovarian blood supply and lower radiation dose) are available [6,7,8,9,10]. The initial reports described small groups of patients and observed a relatively high rate of clinical failure [6,7]. Nonetheless, more recent papers reported more encouraging results and concluded that, in appropriately selected patients, unilateral embolization may have comparable clinical results to standard bilateral UAE [8,9]. In addition to that, Stall et al. who compared unilateral and bilateral UAE observed that unilateral UAE is associated with significantly shorter fluoroscopy time and lower analgesia requirements due to periprocedural pain than bilateral UAE [9]. 

## 2. Objectives

The aim of this study is to present our 10-year experience with unilateral UAE in patients with symptomatic fibroids and to evaluate both radiological and clinical outcomes. 

## 3. Materials and Methods

This retrospective, single-center study was designed to evaluate the outcome of unilateral uterine artery embolization (UAE) in selected patients presenting with symptomatic uterine fibroids. The study was approved by our local institutional review board and was conducted in compliance with the Declaration of Helsinki. Informed consent is not required in our institution for retrospective studies of this type.

The patients included in this study underwent unilateral UAE from January 2010 to December 2021 at a single university center. From a total of 369 UAE procedures, 26 unilateral procedures were identified and further analyzed. All patients underwent preprocedural evaluation during which the treatment options, risks and benefits were discussed with an interventional radiologist and gynecologist. Patient history, baseline symptoms, physical examination, laboratory results and imaging diagnostic (transvaginal ultrasound and magnetic resonance imaging of the pelvis) followed the evaluation. 

## 4. Endovascular Treatment

Prior to the procedure, informed consent was obtained from all patients. In local anesthesia, vascular access was obtained via the femoral or radial artery, and initial angiography was executed using a Pigtail catheter at the level of renal arteries in order to depict the anatomy and the origin of internal iliac and uterine arteries, as well as to exclude potential blood supply from the ovarian arteries. Subsequently, selective catheterization of the uterine arteries was performed using a coaxial system microcatheter placed in the midportion of the uterine artery. The decision of unilateral embolization was made for patients with unilateral fibroid disease supplied from the ipsilateral uterine artery only. In those cases, contributing artery embolization was performed with 700–900 μm calibrated hydrogel microspheres (Embozenes, Varian Medical Systems) based on the uterine artery size and the degree of the blood flow. Control angiography was performed approximately 3–5 min after embolization in order to assure the achievement of an appropriate endpoint. Additional portions of microspheres were injected if needed (Figure 1).

During the procedure, additional IV analgesic treatment with ketoprofen (100 mg), paracetamol (1000 mg) and morphine (up to 12 mg: 3 mg every 15 min) was administered according to the individual needs of the patients. After the procedure, all patients were admitted to the Department of Gynecology for at least a 24 h observation. 

## 5. Follow-Up

All patients returned after 3 to 6 months for a post-procedure evaluation, which included physical examination and assessment of symptoms, as well as control MR imaging of the pelvis. All adverse effects occurring during this period were noted. The re-operation rate was noted. 

## 6. Results

In total, 26 patients met the inclusion criteria. The mean age on admission was 39.7 years (ranging from 28 to 51 years). The chief symptoms included menorrhagia (85%), painful menses (81%), bulk symptoms and difficulty with urination (38%). The reported duration of clinical symptoms was 6.5 months, on average (ranging from 4 to 13 months). 

In all cases, successful embolization was achieved. On average, 2.8 doses (ranging from 1 to 6 doses, 2 mL of particles in each dose) of particles were needed to perform satisfactory vessel occlusion. No intra-procedural complications were noted. In two cases, groin hematoma at the site of the arterial puncture was observed. It did not require surgical intervention. All patients were discharged in good clinical condition. The average hospitalization was 2.7 days (ranging from 2 to 5 days). 

During the follow-up, two patients complained about fever, and two patients reported abdominal pain requiring additional medication. As a minor complication, one patient developed a urinary tract infection, which was treated successfully with oral antibiotics. As far as the re-intervention rate was concerned, one patient underwent repeated embolization, and three were referred for surgical treatment, as no clinical improvement was observed. All procedures were successful. 

During the post-procedural follow-up evaluation, 22 patients (85%) reported at least partial improvement. The patient who required secondary embolization also reported resolution of chief symptoms 6 months after the procedure; hence, the overall success rate was 88%. All patients were asked to express the level of satisfaction on a five-point scale from very satisfied (5) to very dissatisfied (1). Fifteen patients (58%) were very satisfied, seven (27%) were satisfied. Four patients (15%) were dissatisfied with the clinical results. 

The clinical data, procedural details and outcomes are presented in Table 1. 

## 7. Discussion

Traditionally, it has been believed that a successful endovascular treatment of uterine fibroids needs to consist of bilateral uterine artery occlusion. This statement was supported by the observation made by Ravina et al. who described the role of uterine arterial anastomoses in the fibroids’ blood supply [4]. Early experiences with unilateral UAE included small groups of patients but clearly showed a trend of higher re-intervention rate when compared to patients undergoing bilateral vessel occlusion [11]. In addition to this, Gabriel-Cox et al. who evaluated the medical records of 560 patients, 33 of whom had unilateral UAE, identified unilateral UAE as a single risk factor for subsequent hysterectomy [12]. This is why unilateral embolization was considered to be a non-elective, conservative treatment option for cases where technical failure in vessel catheterization occurred, for patients after unilateral uterine artery ligation or, finally, for patients with one-sided anatomical absence of uterine artery [7,8]. 

In 2007, Bratby et al. published an article, which described one center’s experience with UAE. The authors included 48 patients with unilateral UAE among whom the majority (30 cases) was performed as an elective procedure [9]. The decision was made upon angiographic findings in which one uterine artery (smaller and less tortuous) was judged to supply normal myometrium compared to significantly larger and more tortuous artery supplying fibroid tissue. Although the authors reported cases of re-intervention (three in total—two surgical procedures and one repeated UAE), the overall resolution of symptoms was high—from 76% to 85%, depending on the symptoms. The general conclusion was that elective unilateral UAE can achieve positive long-term clinical results in patients with dominant unilateral arterial supply to fibroids. 

A similar study was conducted in 2011 by Stall et al. who reviewed the data of nearly 1500 UAE patients and selected a group of 28 cases with an elective unilateral procedure [10]. The authors compared the procedural findings as well as clinical results of patients who underwent bilateral occlusion and observed few statistically significant differences between the groups. First and foremost, women who underwent unilateral UAE tolerated the procedure significantly better as far as the overall pain and analgesia demand were concerned. Apart from this, a significantly lower dose of radiation was needed if only one uterine artery was embolized. This is a potential advantage of this therapeutic approach, especially considering the fact that UAE patients are usually of child-bearing age, and all measures should be taken to minimize the effects of X-rays on their reproductive organs [13,14]. In terms of clinical results and patient satisfaction, no differences were observed when comparing unilateral and bilateral embolization. Finally, no significant differences were observed in the rate of complete tumor infarction, which is a known factor of a long-term success of endovascular treatment of uterine fibroids [15,16]. 

As far as the overall clinical success and satisfaction rate are concerned, the findings of our study are consistent with observations made by the above-mentioned authors. Clinical improvement was observed in 23/26 (88%) patients, and 85% (22/26) reported to be at least satisfied with the procedural outcome. Nonetheless, the rate of re-intervention (four cases, 15%) seems to be higher than the ones reported by Bratby et al. and Stall et al. This may, however, be attributed to the relatively small sample group. Studies analyzing large groups of patients undergoing endovascular fibroid treatment showed a re-intervention rate ranging from 7% within the first 12 months to 10–25% at 5-year follow-up, which puts the results observed in our study within the acceptable range [17,18]. 

The impact of UAE on pregnancy remains understudied. Older publications reported a high rate of obstetrical complications following the endovascular procedure, including miscarriages, placenta abnormalities and post-partum hemorrhages [19,20,21]. However, more recent studies show that careful and well-planned embolization resulting in ovarian protection (e.g., by using the “fertility-sparing technique” or partial embolization) is a safe and effective procedure for women who want to conceive [22,23]. In our center, all embolizations are performed after thorough evaluation of anatomical conditions in order to minimize the risk of non-target occlusion of ovarian artery. In cases where safe positioning of the micro-catheter distally to the ovarian artery is not possible, larger particles are used. Even though the analysis of the obstetrical outcome was not the aim of the present study, it is a very interesting and important aspect of unilateral uterine artery embolization, and further studies are necessary. 

Significant limitations are present in this series. First and foremost, the retrospective design of the study and the relatively small number of patients limit the validity of the data. Secondly, the absence of a control group treated with bilateral uterine artery embolization might be perceived as a potential drawback. Nonetheless, one should bear in mind that the decision on the therapeutic approach was made after thorough evaluation of the imaging results; hence, a direct comparison of these two groups might not be very relevant. Finally, our study lacks the information on ovarian function after the procedure, which could provide additional information as to whether the risk of ovarian dysfunction is lower or higher among patients undergoing unilateral embolization. We believe that this aspect of unilateral treatment is very interesting, and larger studies are needed.

In conclusion, the results of our study support the statement made by previous authors that unilateral uterine artery embolization might be performed electively in patients with unilateral fibroid disease, as it is associated with a high rate of clinical success, overall patient satisfaction and acceptable rate of re-interventions.

## Figures and Tables

**Figure 1 medicina-58-01732-f001:**
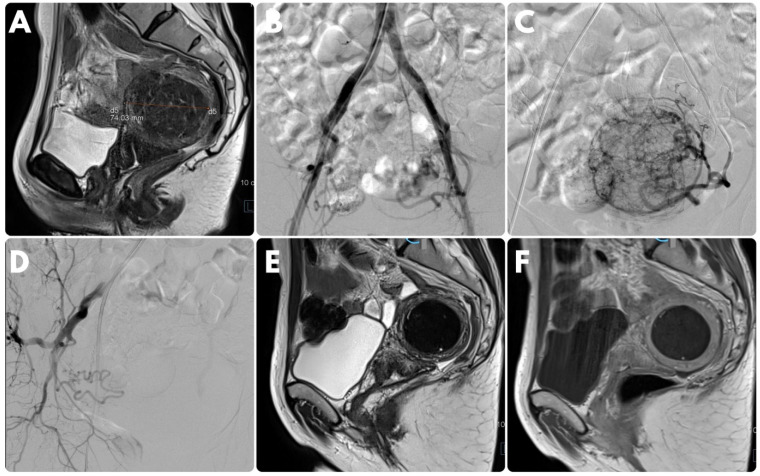
Case of a 43-year-old female patient presenting with menorrhagia and constipation for over a year. Magnetic resonance imaging disclosed the presence of a single posterior intramural myoma (**A**). She was referred for endovascular treatment. Initial angiography performed on the aorta at the level of renal arteries showed that the blood supply to the fibroid originated exclusively from the left uterine artery (**B**). Selective contrast injection performed from the left uterine artery confirmed these findings (**C**). Right uterine artery supplied myometrium (**D**). Complete left-sided unilateral embolization was performed with 700–900 μm particles. Control imaging 3 months after the procedure disclosed significant size reduction in the fibroid (**E**) and no contrast enhancement (**F**). The patient reported noticeable clinical improvement.

**Table 1 medicina-58-01732-t001:** Clinical data, procedural details and outcomes.

**Demographic data**	
Mean age (years, min–max)	39.7 (28 to 51)
Symptoms (*n*, %)	
Menorrhagia	22 (85%)
Dysmenorrhea	21 (81%)
Bulk symptoms	10 (38%)
Difficulty with urination	10 (38%)
Duration of symptoms (months, min–max)	6.5 (4 to 13)
**Procedural details**	
Used embolic material (mL, min–max)	5.6 (2 to 12)
Minor complication rate (*n*, %)	2 (8%)
Post-procedural complication >48 h (*n*, %)	
Fever	2 (8%)
Abdominal pain	2 (8%)
Urinary infection	1 (4%)
Re-intervention rate (*n*, %)	4 (15%)
**Follow-up**	
Clinical improvement (*n*, %)	23 (88%)
Overall satisfaction (*n*, %)	
Very satisfied	15 (58%)
Satisfied	7 (27%)
Neutral	0 (0%)
Dissatisfied	4 (15%)
Very dissatisfied	0 (0%)

## Data Availability

Supporting data is available on request—mszmygin@gmail.com.

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
