# Peer review of "Unilateral Uterine Artery Embolization as a Treatment for Patients with Symptomatic Fibroids—Experience in a Case Series"

_medicina, 2022, doi:10.3390/medicina58121732_

Round 1
Reviewer 1 Report
The manuscript sent for review concerns a very interesting problem of conservative treatment of uterine fibroids. Personally, I am a supporter of surgical treatment, but all reports on minimally invasive methods are interesting.
My comments:
The affiliations of the 3 authors are the same and appear 3 times
What the authors mean, for example in the abstract, is that fibroids were symptomatic? - this needs to be supplemented.
The unexplained abbreviation UFE appears in the abstract
It needs to be clarified that the group of 369 women from the UAE was examined and only 26 women were included in the analysis.
The situation in which embolization is performed before a planned pregnancy also requires clarification. Isn't that contraindicated? Literature analysis is needed.
I think this is a very interesting study and should be published with minor adjustments.
Author Response
Dear Reviewer,
Thank You very much for You review and constructive comments. We tried to address all of them and incorporate Your remarks into our manuscript. We hope that the revised version of the manuscript might be published in present form. Should You have any further comments, do not hesitate to write them.
Kind regards
Maciej Szmygin
The affiliations of the 3 authors are the same and appear 3 times
Indeed, yet in all 3 cases they differ in terms of some details: 1st author is not a corresponding author, 2nd author is a corresponding author and therefore all the contact details are included and the 5th author is the Head of the Department.
What the authors mean, for example in the abstract, is that fibroids were symptomatic? - this needs to be supplemented.
Explanation was added.
The unexplained abbreviation UFE appears in the abstract
The abbreviation was replaced by UAE.
It needs to be clarified that the group of 369 women from the UAE was examined and only 26 women were included in the analysis.
The statement was added both in abstract and Materials and Methods section.
The situation in which embolization is performed before a planned pregnancy also requires clarification. Isn't that contraindicated? Literature analysis is needed.
Thank You for this very interesting remark – we added a paragraph with a literature analysis concerning pregnancy after UAE to the Discussion.
I think this is a very interesting study and should be published with minor adjustments.
Reviewer 2 Report
The manuscript is very interesting, dealing with a much debated topic among experts.
Unilateral embolization of the uterine arteries for hemodynamic conditions, technical difficulties, anatomical variants and unilateral dominance of blood supply to the fibroid.
Certainly 26 patients are not many, so I would suggest changing the title by inserting, after the title.....: EXPERIENCE ON A CASE SERIES.
In M&Ms it would be useful to know what kind of 700–900 μm micro-spheres the authors used.
What the authors mean when they write: Additional embolic material was injected if needed. Did they use Spongostan or other micro-spheres or micro-coils?
A general text check would also be useful to correct minor typographical and editing errors.
Author Response
Dear Reviewer,
Thank You very much for You review and constructive comments. We tried to address all of them and incorporate Your remarks into our manuscript. We hope that the revised version of the manuscript might be published in present form. Should You have any further comments, do not hesitate to write them.
Kind regards
Maciej Szmygin
The manuscript is very interesting, dealing with a much-debated topic among experts.
Unilateral embolization of the uterine arteries for hemodynamic conditions, technical difficulties, anatomical variants and unilateral dominance of blood supply to the fibroid.
Certainly 26 patients are not many, so I would suggest changing the title by inserting, after the title.....: EXPERIENCE ON A CASE SERIES.
The title was modified accordingly.
In M&Ms it would be useful to know what kind of 700–900 μm micro-spheres the authors used.
The explanation (700–900 μm calibrated hydrogel microspheres (Embozenes, Varian Medical Sys-tems) was added to the section.
What the authors mean when they write: Additional embolic material was injected if needed. Did they use Spongostan or other micro-spheres or micro-coils?
Indeed, this statement was misleading – we corrected it – “Additional portions of microspheres were injected if needed”
A general text check would also be useful to correct minor typographical and editing errors.
The text was checked and corrected.